# Extending the Shelf-Life of Immunoassay-Based Microfluidic Chips through Freeze-Drying Sublimation Techniques

**DOI:** 10.3390/s23208524

**Published:** 2023-10-17

**Authors:** Sangjun Moon

**Affiliations:** 1Department of Mechanical Convergence Engineering, Gyeongsang National University, Changwon 51391, Gyeongsangnam-do, Republic of Korea; nanobiomems@gnu.ac.kr; Tel.: +82-55-250-7304; Fax: +82-55-250-7399; 2Cybernetics Imaging Systems Co., Ltd., Changwon 51391, Gyeongsangnam-do, Republic of Korea; 3Department of Mechanical Engineering, Ulsan National Institute of Science and Technology (UNIST), Ulsan 44919, Republic of Korea

**Keywords:** shelf-life extension, immunoassay chip, microfluidics, freeze-drying, sublimation

## Abstract

Point-of-care testing (POCT) platforms utilizing immunoassay-based microfluidic chips offer a robust and specific method for detecting target antibodies, demonstrating a wide range of applications in various medical and research settings. Despite their versatility and specificity, the adoption of these immunoassay chips in POCT has been limited by their short shelf-life in liquid environments, attributed to the degradation of immobilized antibodies. This technical limitation presents a barrier, particularly for resource-limited settings where long-term storage and functionality are critical. To address this challenge, we introduce a novel freeze-dry sublimation process aimed at extending the shelf-life of these microfluidic chips without compromising their functional integrity. This study elaborates on the mechanisms by which freeze-drying preserves the bioactivity of the immobilized antibodies, thereby maintaining the chip’s performance over an extended period. Our findings reveal significant shelf-life extension, making it possible for these POCT platforms to be more widely adopted and practically applied, especially in settings with limited resources. This research paves the way for more accessible, long-lasting, and effective POCT solutions, breaking down previous barriers to adoption and application.

## 1. Introduction

The development of surface modification techniques has significantly advanced the field of microfluidic platforms [1], enabling the immobilization of antibodies on various substrates such as polymer [2,3,4,5], silicon [6], and glass for the purpose of capturing specific cells and proteins [7,8]. Particularly in point-of-care testing (POCT) applications, immunoassay-based surface chemistry has emerged as a versatile and highly specific method [9], leveraging solid surfaces that are conjugated with target antibodies to facilitate detection [10,11,12,13,14,15]. To further enhance surface functionality for POCT [16], nanoparticles have been employed as solid matrices for antibody immobilization [17,18,19,20]. However, this approach presents challenges for deployment in resource-limited settings, where the immunoassay-based microfluidic chips are subject to degradation over time [21]. Long-term preservation of chip functionality becomes even more critical in such settings, given the challenging field conditions for POCT, including high temperatures, moisture exposure, and prolonged storage times. Moreover, these advanced diagnostic technologies need to be adaptable to the medical testing environment in developing countries, which often lack the infrastructure commonly available in well-resourced settings. Traditional medical equipment often necessitates a plethora of resources: climate-controlled laboratories, refrigerated reagent storage, stable electrical power, highly trained personnel, rapid sample transportation, and a constant supply of calibrators to maintain equipment performance [22]. Therefore, it is imperative to develop methods for extending the shelf-life of immunoassay-based microfluidic chips, making them suitable for resource-limited settings without compromising their diagnostic capabilities [23,24].

Simplifying immunoassay-based microfluidic platforms has been a focus of recent advancements [25,26], aiming to provide user-friendly solutions with minimal steps required for sample preparation, particularly for point-of-care testing (POCT) applications [27,28,29,30,31]. These platforms have demonstrated their versatility by selectively capturing HIV particles [28], CD4^+^ cells [27,30], and even isolating CD4^+^ monocytes from whole blood in resource-limited settings [29]. However, a significant hurdle impeding the widespread adoption of POCT is the instability of conventional surface chemistry, which usually requires a liquid environment to maintain functionality. Biological materials, on the other hand, often need to be dried for long-term storage and distribution. Due to this conflicting requirement, the liquid-based immunoassay systems face challenges related to the degradation of immobilized antibodies, limiting their shelf-life and subsequent application in POCT settings. Therefore, a new approach tailored for resource-constrained environments must be introduced to overcome this technological barrier and preserve the functionality of microfluidic-based POCT systems. This necessitates the development of an all-in-one dry chemistry microfluidic chip for robust POCT applications [32]. Protein conformations and antibody functionalities are highly sensitive to their surrounding environments [13,33]. As a result, the surface chemistry in the channels of these devices tends to degrade over time, particularly when exposed to harsh conditions like high temperatures and high humidity, common in tropical regions where they are frequently utilized for POCT [34].

In this work, we introduce a freeze-drying sublimation process designed to mitigate the degradation of immobilized antibodies, thereby extending the shelf-life of immunoassay-based microfluidic chips. Freeze-drying, also known as lyophilization or cryo-desiccation, is a preservation method that involves controlled freezing followed by the sublimation of ice under a vacuum. This technique has been traditionally employed across various industries such as pharmaceuticals, food, and biotechnology to preserve temperature-sensitive materials like bovine serum albumin (BSA) [32], food products [33], and antibodies [31]. However, applying freeze-drying methods to microfluidic devices presents unique challenges, particularly concerning the impact on materials used for device fabrication, such as PMMA, glass slides, and double-sided adhesive films. Moreover, it also influences the functionality of chemicals and reagents used for surface modification, such as blocking solution BSA and 3-Mercaptopropyl-trimethoxysilane, among others. In prior work [35,36,37], Ramachandran et al. utilized lyophilization to initially dry the reagents before storing them on the microfluidic chip surface, enabling prolonged preservation while maintaining device functionality. Their device was capable of pathogen immunocapture as well as PCR-based nucleic acid amplification for pathogen identification. While this approach leverages surface fixation to matrices, applying it directly to the sensitive planar micro-fixation technique aimed at detecting specific cells within blood can be challenging. Furthermore, by using alternative reagents based on sugars like trehalose, it necessitates modifying the surface using a cleaning solution before introducing the blood sample. These additional steps can impact the capture efficiency of specific cells, demanding preservation methods for equipment at the end-user stage. In our approach, we have developed a specialized lyophilization protocol for preserving each of the various reagents utilized in our microfluidic device, including antibody-coated magnetic beads, lysis buffer, and PCR mix. This individualized freeze-drying method is specific to the microfluidic chip and involves considerable time and effort to optimize. Once freeze-dried, these materials are sealed to prevent degradation from moisture, facilitating extended preservation and subsequent reconstitution when required. Therefore, each microfluidic device employing this technology will necessitate its own uniquely tailored preservation strategy based on the specific reagents in use. Our findings thus offer a pioneering technique for extending the utility of immunoassay-based microfluidic systems, especially in resource-limited settings, making them more adaptable and durable for long-term applications.

In the realm of point-of-care (POC) technology, particularly in the monitoring of HIV, preserving the functionality of microfluidic chips over time is of paramount importance. Conventional methods of preserving these chips often lead to the degradation of their functionality, substantially reducing their ability to efficiently capture cells like CD4^+^ T cells. Therefore, we propose a novel approach to maintaining the functionality of these chips through freeze-drying, also known as lyophilization. While freeze-drying methods have been employed extensively in various industries to preserve biological materials, the application of this technique to microfluidic devices containing immobilized proteins and antibodies is unprecedented. In this study, we specifically focus on freeze-drying immunoassay-based microfluidic devices designed to capture and count CD4^+^ cells, a crucial component in POC HIV monitoring. Our primary objective is to extend the shelf-life of these devices by preserving the functional integrity of the immobilized antibodies. These freeze-dried microfluidic devices can subsequently be sealed for long-term storage or transportation and reconstituted when needed, without compromising their efficacy. Our innovative freeze-drying process involves immersing the entire microfluidic chip in liquid nitrogen, followed by lyophilization under low pressure to sublime the immobilized antibody directly from a solid to a gaseous state. This method not only maintains the active state of the antibody but also significantly enhances the overall shelf-life of the chip. As a result, we adapt this drying process to antibody-immobilized POC devices and evaluate its effectiveness in capturing CD4^+^ cells using blood samples from HIV patients. In terms of the technical setup, the simplest lyophilizer would consist of a vacuum chamber for the wet sample, equipped with a mechanism to remove water vapor, thus initiating evaporative cooling and eventual freezing as shown in Figure 1. To extend the shelf-life of a microfluidic chip designed for POCT, a freeze-drying technique is applied. The chip, initially functionalized in a liquid environment for immunoassay procedures, needs to be freeze-dried to maintain its effectiveness until used for blood testing. Figure 1b shows the fabrication workflow: The freeze-drying process involves a sublimation phenomenon under low-pressure conditions to convert the liquid phase into a solid. These steps include (i) immersing the antibody-immobilized microfluidic chip in liquid nitrogen to solidify the buffer solution; (ii) reducing ambient pressure while maintaining low temperature, without changing the phase; and (iii) initiating sublimation by incrementally increasing pressure, thus turning the solid phase directly into the gaseous phase without transitioning through a liquid state.

Recent advancements in POC technology have led to the development of immunoassay-based microfluidic devices capable of selectively capturing CD4^+^ cells from the whole blood of HIV-infected patients with high specificity and efficiency. These devices are especially crucial for resource-limited settings, where conventional medical equipment may not be readily available. Freeze-drying our devices offers multiple advantages: it extends the shelf-life, eliminates additional refrigeration costs, and makes them more robust and easier to handle under the high-temperature and moisture-rich conditions commonly found in tropical countries. This, in turn, makes these devices highly suitable for POC applications in resource-limited settings. To quantify the effectiveness of our freeze-drying technique, we compared the cell capturing efficiency of the freeze-dried chips with that of regular, non-freeze-dried chips. Our findings demonstrate not only the feasibility but also the enhanced performance and adaptability of our freeze-dried immunoassay-based microfluidic CD4^+^ T cell counters for POC applications, especially in settings with limited resources. Thus, our study contributes a groundbreaking method in the field of POC technology, potentially revolutionizing the way we diagnose and monitor diseases like HIV.

## 2. Materials and Methods

### 2.1. Sublimation Model Analysis

The numerical simulation of the freeze-drying process for the engineered microfluidic chip is conducted using COMSOL Multiphysics^®^ modeling software, V5.0. This software allows us to perform coupled physics calculations that incorporate heat flux, diffusion, and the equilibrium between ice and vapor. We configured the model based on the conditions of ideal gas and pure water. As illustrated in Figure 2a, the model consists of two physical domains—solid ice and vaporized air—governed by separate equations and interconnected through a thermodynamically balanced ice–vapor interface.

The heat conduction equation for both the ice and vapor phases can be described as:(1)$$\rho C_{p}\frac{\partial T}{\partial t}=\nabla \cdot k\left(\nabla T\right)\;$$

Here, ρ, $C_{p}$, and k represent the density, specific heat capacity, and thermal conductivity for both ice and vapor, respectively. This equation outlines how changes in heat energy alter temperature distribution, causing phase changes and affecting density and vapor pressure at the interface.

When the ice transitions to its vapor phase, mass transfer occurs due to concentration gradients at the surface. This can be formulated using a mass transfer equation:(2)$$\frac{\partial c}{\partial t}=\nabla \cdot \left(D\nabla c\right)\;$$

In this equation, c and D symbolize the concentration of mass and diffusion coefficient, respectively. As the ice melts, the ice–vapor interface moves downward, ultimately leaving only the vapor phase due to dynamic equilibrium phenomena.

At this interface, thermodynamic equilibrium and mass balance are represented by the following equations:(3)$$T=\frac{P_{eq}}{Rc}\;$$
(4)$$\;V_{s}=-\frac{Q_{s}}{\rho_{ice}L}\;$$

Here, $P_{eq}$ and R are the equilibrium vapor pressure and gas constant, respectively. The interface velocity,$\;V_{s}$, can be calculated using the normal heat flux $Q_{s}$, and the latent heat of sublimation L.

Upon applying these equations, numerical simulations revealed the morphological changes at the interface and the total freeze-drying process time. As illustrated in Figure 2c and Appendix A, the complete freeze-drying procedure took approximately 12,900 s or 3 h and 51 min, with the interface demonstrating both circular and linear shapes as it moved downward.

### 2.2. Microfluidic Chip Fabrication

The microfluidic chip was fabricated in under a minute per unit, avoiding costly lithographic techniques. Utilizing CAD software (AutoCAD V18.0) to operate a laser cutter, rapid prototyping of the device was achieved. The primary materials used were a 3.175 mm-thick clear cast acrylic sheet (PMMA) and a 50 µm-thick double-sided adhesive film (DSA, 3M 8142). These materials were machined with a VersaLASER Platform equipped with a 30 W CO_2_ laser, creating 0.75 mm diameter inlets and outlets along with 24 mm-long by 4 mm-wide fluidic channels. The PMMA and adhesive film were manually aligned with a 250 µm tolerance and bonded together after removing the protective backing. This PMMA-DSA assembly was then immediately sealed with a cover glass slip, which had been freshly exposed to air plasma via a plasma generator.

### 2.3. Chemical Reagents Preparation

3-Mercaptopropyl trimethoxysilane (3-MPS) was sourced from Gelest, while ethanol and glass slides were obtained from Fisher Scientific. Dimethyl sulfoxide (DMSO), lyophilized bovine serum albumin (BSA), isopropanol, and a glove-bag for moisture-sensitive silane handling were procured from Aldrich Chemical Co., Boston, MA, USA. The coupling agent GMBS, along with NeutrAvidin™ biotin-binding protein, were acquired from Pierce Biotechnology. Gibco supplied 1× Phosphate Buffered Saline (PBS), and biotinylated mouse anti-human anti-CD4 was purchased from Beckman Coulter. Additional reagents such as DAPI, Alexa Fluor^®^488 and Alexa Fluor^®^647 antibodies, and BD FACS Lysing Solution were sourced from Invitrogen and BD Bioscience, respectively.

We prepared a 4% (*v*/*v*) 3-MPS solution in ethanol for silanization within a nitrogen gas-filled glove bag. A GMBS stock solution was formulated with 50 mg GMBS dissolved in 0.5 mL of DMSO, and later diluted in ethanol. NeutrAvidin™ solution was reconstituted according to the manufacturer’s guidelines using 10 mg in 1 mL PBS. Additionally, a 10% (*v*/*v*) solution of biotinylated mouse anti-human anti-CD4 was prepared in PBS containing 1% (*w*/*v*) BSA to create the CD4 antibody solution. Finally, a 10% (*v*/*v*) BD FACS Lysing solution in PBS was mixed to formulate the cell-fixing solution.

### 2.4. Antibody Immobilization on the Channel Surface

Freshly created microfluidic devices are treated with specialized surface chemistry to effectively capture CD4^+^ cells when a blood sample is injected. Initially, glass slides are pretreated with a salinization solution for an hour at room temperature, prior to being joined to the PMMA base. The inner surface of the channel then undergoes a 60 min incubation at room temperature with a GMBS solution. This is followed by a 60 min incubation with NeutrAvidin™ at 4 °C. The final step involves a 30 min reaction with anti-CD4 antibody solution at 4 °C, which is then re-injected to ensure uniform antibody distribution, thus enhancing cell capture efficiency. Covalent bonds form between functional groups’ exposed side chains, resulting in irreversible bonds and high surface coverage, as illustrated in Figure 2b.

Each step of surface modification is followed by a cleaning process involving ethanol or PBS, based on the solvent used in the preceding step. A comprehensive surface chemistry protocol is outlined in Appendix A. Once surface-treated, these devices can be stored for up to a month in a refrigerator, sealed with PARAFILM^®^, Boston, MA, USA.

### 2.5. Freeze-Drying Process

After the surface-treated microfluidic chips are incubated with biotinylated anti-CD4 antibody, they are prepared for preservation through freeze-drying. Chips processed up to the NeutrAvidin™ or anti-CD4 antibody immobilization stage are placed in 55 mm Petri dishes and submerged in liquid nitrogen for 5 min. They are then freeze-dried for 24 h at room temperature using a bench-top freeze-drying system (Freezone^®^ 4.5 LABCONCO, Boston, MA, USA) at a 1.2 Pa vacuum, as depicted in Figure 1b. Due to differential expansion coefficients between PMMA and glass, the chips may partially detach during freezing; they are reassembled by pressing the PMMA cover back onto the cover glass. These freeze-dried chips are sealed with PARAFILM^®^ for future use. A detailed freeze-drying protocol is available in Table 1.

### 2.6. Functional Surface Evaluation

Surface functionality at each stage was assessed using fluorescently tagged biomolecules: biotin-FITC and soluble anti-CD4-FITC (Pierce, Boston, MA, USA). After treating with fluorescent stains, devices were rinsed with 100 µL of PBS to remove unbound stain. Imaging was carried out using a Zeiss automated microscope, Boston, MA, USA at 10× magnification.

### 2.7. Blood Injection and Target Cell Capture Experiment

In the methodology, 4 µL of buffy coat blood samples were introduced into the custom microfluidic channel, which was surface-treated for CD4 antigen capture, employing both conventional chemistry and freeze-drying processes. The sample was injected via pipette tips into the inlet, sized to ensure that captured cells would sufficiently cover the channel floor. Following injection, a 60 µL PBS wash solution was immediately added using the same inlet. The wash fluid was then allowed to flow at 20 µL/min for 3 min to remove any unbound cells, including red blood cells and undesired CD3-CD4^+^ monocytes, through shear force. The injection and wash flow rates were determined based on previous analytical results and executed with a low auto-pipette.

### 2.8. Captured Cell Staining and Imaging

To evaluate the functionality of the device, specifically capture specificity and efficiency, captured cells were fixed with a cell-fixing solution for 15 min. Staining solutions were prepared as follows: a mixture of 100 μL AF488-anti-CD4, 100 μL AF647-anti-CD3, and 800 μL PBS was used for fluorescent antibody binding. Additionally, a DAPI staining solution combined 1 μL of DAPI stock (5 mg/mL) with 1000 μL PBS. These solutions were introduced into the channel and incubated 30 min at room temperature for antibody binding and 90 min at 4 °C for enhanced fluorescence.

Subsequent to PBS washing to remove excess staining, cells were visualized with a fluorescence microscope. Imaging was conducted using a 1 mm^2^ field of view (FOV) and a 10× objective lens. A total of four measurements were made using various fluorescence filters, capturing 384 adjacent images that cover the entire channel (4 in width × 24 in length × 4 for different fluorescent colors). Imaging was accomplished using four different color filter tubes: UV (359 nm/461 nm) for DAPI, GFP (489 nm/509 nm) for AF488, and Cy5 (650 nm/670 nm) for AF647, each corresponding to specific excitation and emission wavelengths.

Different filter sets were used for regular and freeze-dried devices. DAPI staining (blue) was employed to differentiate red blood cells from other artifacts. Green and red fluorescent stains, corresponding to AF488 and AF647, were used to distinguish between CD4^+^ monocytes and CD3^+^CD4^+^ T-lymphocytes, essential for disease monitoring.

### 2.9. Cell Counting and Evaluation of Cell Capture Efficiency

Post-imaging, total cell counts for DAPI, GFP, and Cy5 images are determined using specialized lab-developed software, V1.0. Each image is segmented into seven regions of interest (ROIs) for precise cell counting. To assess the effectiveness of the new freeze-drying-based surface chemistry against traditional methods, a first-order polynomial regression is employed to calculate R-squared (R^2^) values. Capture efficiency for the freeze-drying process is estimated by comparing the total cell counts obtained from the microfluidic device.

## 3. Results

### 3.1. Fabrication Results

A freeze-dried immunoassay microfluidic chip was developed to extend the functional surface’s shelf-life. The fabrication involves a sublimation phenomenon, transitioning the liquid to a solid in a low-pressure environment, as illustrated in Figure 1b. The process unfolds in three phase-changing steps. Initially, a microfluidic chip with immobilized antibodies is dipped in liquid nitrogen, converting the buffer solution from liquid to solid. Subsequently, the chip’s environment is exposed to a vacuum to maintain low temperature without altering its phase. Finally, an increase in pressure triggers sublimation, turning the solid directly into gas without transitioning to liquid, as depicted in Figure 1b(i–iii).

The microfluidic chip is crafted using a double-sided adhesive-based device, produced through CO_2_ laser cutting and manual bonding, as elaborated in the Section 2 and Figure 2a(i). Channel geometry is simulated using COMSOL Multiphysics^®^ software, V5.0, in line with our prior work. Subsequently, the chips undergo regular surface chemistry, as seen in Figure 2a(ii). For freeze-drying, chips are immersed in liquid nitrogen and then placed in airtight containers connected to a benchtop freeze-drying system, forming a vacuum. This changes the surface to a lyophilized state, displayed in Figure 2a(iii).

Post lyophilization, porous structures remain, allowing the chips to be sealed and stored at atmospheric conditions without degrading the immobilized antibodies. When needed for testing, a PBS buffer rehydrates the surface, and whole blood is introduced to capture target cells, as illustrated in Figure 2a(iv). The porous structure left after sublimation eases device reconstitution, as capillary forces enable easy fluid filling.

The surface chemistry involves four primary steps to attach target antibodies to a glass slide, as illustrated in Figure 2b. Post-plasma treatment, silane-based self-assembly is conducted on the glass slide (Figure 2b(i)). Subsequently, GMBS is immobilized using thiol-maleimide coupling (Figure 2b(ii)). Thiol groups form an addition reaction with the maleimide double bond. NeutrAvidin™ is then conjugated to the surface through NHS ester bonding to amplify specific binding affinity with biotin molecules (Figure 2b(iii)). Lastly, biotin-conjugated antibodies interact with NeutrAvidin™ via protein affinity binding (Figure 2b(iv)).

For the modified chip, sublimation is modeled as described in the Section 2 and displayed in Figure 2c(i). Using COMSOL Multiphysics^®^ software V5.0, channel geometry and sublimation process parameters are simulated, as revealed in Figure 2c(ii) and Appendix A. Simulations depict how the liquid–solid interface adapts to channel geometry and how this interface lags during the solid-to-gas phase transition, taking up to 4 h. This calculated time frame informs the experimental protocol for benchtop freeze-drying equipment.

### 3.2. Surface Function Evaluation

The evaluation of chip surface functionality is executed through fluorescence tagging at each stage of surface chemistry, as depicted in Figure 3. Fluorescently tagged biomolecules, namely biotin-FITC and soluble anti-CD4-FITC (Pierce, MA), are used to assess the functions of the freeze-dried surface. Post-incubation with these fluorescent stains, the devices are rinsed using 100 µL of PBS to clear residual staining and fixing agents. Zeiss automated microscopy at 10× magnification captures images at five points, spaced 5 mm apart from the inlet (Figure 3a).

Relative fluorescent intensity is quantified against background levels, specifically contrasting the channel interior and exterior. For three NeutrAvidin™ and biotin-FITC concentrations (5, 10, and 20 μg/mL), intensity is gauged at five different locations (Figure 3b). Values are normalized to the maximal intensity of 3500, observed at spot one with a 20 μg/mL concentration combination of NeutrAvidin™ and biotin-FITC. Intensity variances across the chip surface are marginal, with standard deviations ranging from 0.01 to 0.07, except at high concentration levels (20 μg/mL), where the standard deviations are 0.1 and 0.12. At such high concentrations, intensity dwindles from inlet to outlet due to surface conjugation saturation of the target molecule, biotin-FITC, with pre-immobilized NeutrAvidin™. Furthermore, the differences in fluorescence occurring at each spot can be attributed to the surface forces resulting from shear stress distribution in the fluid where laminar flow occurs, as well as the variables of diffusion (both temporal and spatial) from physical and chemical absorption during chemical reactions.

Two-way ANOVA without replication reveals significant differences related to both conjugation molecules and their concentrations, with *p*-values of 0.04 for NeutrAvidin™ and 0.001 for biotin—falling below the 95% confidence level (α = 0.05). This suggests that biotin concentration primarily influences the effectiveness of surface chemistry. For instance, a doubling of biotin-FITC concentration from 10 to 20 μg/mL yields a tripling of total fluorescence intensity, indicating abundant binding sites conforming to its 1:4 biotin-to-avidin pockets ratio. Normalized values shift accordingly: from 0.27 to 0.81 for 20 μg/mL and 0.13 to 0.69 for 10 μg/mL of NeutrAvidin™. Overall, high concentrations of both avidin (≥10 μg/mL) and biotin (≥20 μg/mL) do not substantially alter the freeze-dried surface’s functional efficiency in NeutrAvidin-Biotin-based immunoassays.

### 3.3. Comparing Normal and Freeze-Dried Chips

After assessing NeutrAvidin and NeutrAvidin-biotin binding efficiency, the focus shifts to utilizing biotinylated antibodies to evaluate immunoassay functionality. A comparative analysis is performed between normal and freeze-dried chips using whole blood injections (Figure 4a). The freeze-drying process begins by immersing the surface-modified chip in liquid nitrogen (Figure 4a(i)) and then reassembling the chip through gentle pressing (Figure 4a(ii)). These chips can be stored at room temperature and are vacuum-sealed using a cost-effective sealer to maintain their integrity (H-1075, Uline, Allentown, PA, USA).

When testing is needed, whole buffy coat blood is injected post-rehydration with a buffer solution (Figure 4a(iii)). Target CD4^+^ cells are captured during this stage using a controlled flow rate. Subsequently, non-adherent whole blood is washed off (Figure 4a(iv)). Captured cell numbers are then analyzed through fluorescent and bright-field imaging via specialized cell counting software. Comparative fluorescent imaging is conducted between non-freeze-dried and freeze-dried chips (Figure 4b,c).

In normal chips, overlay images for DAPI/AF488/AF647 are represented in Figure 4b(i). These images distinguish between cell types and artifacts using different biomarkers—DAPI for nuclei (Figure 4b(ii)), AF488 for CD4^+^ cells (Figure 4b(iii)), and AF647 for CD3^+^ cells (Figure 4b(iv)). Based on this overlay, only cells positive for both CD4 and CD3 are considered targets and used to evaluate the surface chemistry’s efficacy in capturing them.

For freeze-dried chips, similar fluorescent and bright-field images are acquired using DAPI, AF488, and AF647 (Figure 4c(ii–iv)). Both chip types display different efficiencies and specificities in capturing target cells, defined by ratios of DAPI counts to bright-field and fluorescent dots (Figure 4d(i,ii)). Counts are taken at seven different points along the chip’s length (Figure 4d(i,ii)).

Overall, both chip types show a similar capture pattern based on DAPI/AF488/AF647 counts (Figure 4d(iii)). Efficiency is normalized using bright-field images from six identical chips with the same blood sample. The capture patterns, average counts, and standard deviations between the two chip types are not significantly different when normalized to the total number of captured cells. Statistical analysis using the Student’s *t*-test indicates no significant difference between the chips (*p* = 0.208), exceeding the 95% confidence level threshold (α = 0.05).

Lastly, standard deviation values indicate a mere 10% difference between the two types, with values of 0.43 and 0.33, respectively. This underscores the comparability of the two chip types in terms of target cell capture efficiency and stability, validating the efficacy of the freeze-dried surface chemistry in immunoassay applications.

### 3.4. Comparing Three Microfluidic Chips with Blood

A performance comparison among three types of microfluidic chips—normal, freeze-dried, and a commercially available non-specific protein chip—is depicted in Figure 5a. Each chip type was evaluated using three different blood samples, yielding a dataset with three channels per chip and three blood samples. The commercial chip employs an avidin-coated glass substrate sourced from Xenopore Slide Glass Company and requires multiple chemical steps for surface modification.

Cell counting was performed using a lens-less holographic imaging platform. Results indicate negligible differences between normal and freeze-dried chips when employing full chemical linkage for antibody immobilization. The average maximum and minimum cell count deviations are 40 ± 125 (15.7%) and 100 ± 65 (5.8%), respectively. In contrast, the commercial chip shows wider variations—227 ± 104 (35.6%) and 119 ± 134 (17.7%). A two-way ANOVA analysis without replication corroborates the findings, revealing no significant variance in performance between chips when tested with normal blood (*p* * = 0.43). However, chips display a statistically significant difference in surface immobilization methods (*p* ** = 0.004), under the 95% confidence level (α = 0.05).

Correlation analysis is presented in Figure 5b, examining 12 and 18 microfluidic chips using two different blood samples (Figure 5b(i,ii)). Employing a first-order regression curve, high correlation coefficients of 0.97 and 0.93 are calculated for the two different blood samples. F-test values confirm the correlation, yielding *p*-values of 0.000032 and 0.000046, indicating a strong fit with the first-order regression models. Further analysis via Student’s *t*-test, based on two-sided distributions with equal variance, also supports the data, with *p*-values of 0.763 and 0.664 for the two blood samples, exceeding the 95% confidence threshold (α = 0.05).

In summary, freeze-dried chips demonstrate comparable cell capture efficacy to regular chips, even across different blood samples. This affirms the freeze-drying approach as a feasible, alternative storage and utilization method for microfluidic applications.

## 4. Discussion and Conclusions

Different material properties in microfluidic chips present challenges during freeze-drying and rehydration, particularly the varying shrinkage rates of materials like PMMA, double-sided adhesive film, and glass. These materials possess unique coefficients of thermal expansion, leading to issues like bonding failure during rapid immersion in liquid nitrogen. Despite these challenges, successful bonding is achieved between the upper PMMA port and lower glass slide using a double-sided adhesive film, which is backed by polyimide. This backing material exhibits a complex, entangled microstructure that helps maintain a weaker but sufficient bond between surfaces.

Another approach to avoid liquid nitrogen in the sublimation process is the use of a laboratory lyophilizer. In this method, samples are frozen in a flask connected to an ice condenser, which eliminates the need for manual handling that could result in component detachment. After drying, vacuum sealing is necessary to eliminate humidity changes during chip transport.

As for the chip’s long-term performance, tests indicate that freeze-dried chips maintain comparable cell capture efficiency to regular chips even after 10 days of vacuum-sealed storage. Therefore, these findings suggest that despite the material challenges associated with freeze-drying and rehydration, effective bonding and cell capture can still be achieved, providing viable alternative methods for chip storage and use.

Following the sublimation process, residual white salt particles were observed within the microfluidic channels, consisting of unbound antibodies and salts from the PBS buffer solution. These particles could interfere with the functionality and specificity of the chip by altering the pH and affecting antibody activities. To address this, a two-step washing process using PBS was employed to remove these particles, which had no adverse effect on the chip’s performance, as corroborated by the Section 3.

Further benefits of this rehydration process include the precise dispensing of the reinjection buffer and the option to sterile filter it immediately before transferring to final storage, reducing the risk of contamination from particles and bacteria. Experimental setups utilized various flow rates for injecting buffy coat blood samples to evaluate the CD4^+^ cell capture efficacy of these freeze-dried chips. A 5 µL blood sample was injected using an auto-pipette at a consistent speed, followed by a 5 µL PBS wash to remove unbound RBCs and CD4^+^ monocytes. This was succeeded by fixation using a 100 µL FACS lysing solution, and fluorescent staining was applied to differentiate captured CD4^+^ T cells from monocytes.

Contrary to expectations, neither the additional fluorescent staining steps nor the rate of auto-pipette injection had any discernible impact on the surface functionality when compared to chips with standard surface chemistry. In-line holographic imaging was employed for cell counting, eliminating the need for fluorescence tagging. An adaptive cross-correlation method was used in the counting algorithm, negating the need for extra manpower to assess the number of captured cells. Notably, the performance of the freeze-dried chips was in agreement with flow cytometry results, indicating their potential to resolve point-of-care (POC) challenges associated with refrigeration and drying.

In point-of-care testing (POCT) with microfluidic platforms, one might consider simply drying proteins onto a chip surface as an attractive alternative, as this method sidesteps complex surface chemistry processes. These simpler drying techniques may even offer easier bio-preservation compared to our proposed freeze-drying method. However, our experimental data, along with other studies, suggest that for reliable and repeatable capture efficiency, as well as improved sensitivity and specificity, the orientation of antibodies on the microchip surface is critical.

Simpler methods like physisorption fail to provide the architectural consistency required for effective performance. Moreover, antibodies attached via physisorption exhibit low adherence and are at risk of being washed away when biofluids are introduced into the microchip. As our results indicate, more robust surface chemistries are essential for these microchips to operate within clinically acceptable error margins. Therefore, our freeze-drying approach emerges as a viable solution for preserving complex surface chemistries, eliminating the need for refrigeration, and extending shelf-life in point-of-care settings.

In conclusion, microfluidic chips with immunoassay-based surface chemistry offer specificity and versatility across various applications. However, their usage in point-of-care testing (POCT) has been limited by the shelf-life concerns arising from the degradation of immobilized antibodies in a liquid environment. To address this issue, especially in resource-limited settings, we introduced a freeze-drying sublimation process aimed at preserving the chip’s functionality over an extended period.

Our study compared the cell-capture efficiency of freeze-dried chips with that of regular, liquid-based chips. Our findings show no significant difference in functionality between the two types, with a 95% confidence level. Importantly, the cell capture efficiency and specificity of the microfluidic channels remained consistent post-freeze-drying. This congruence between the freeze-dried and the liquid-based chips indicates the viability of the freeze-drying approach as a solution to the challenges of refrigeration and long-term storage in POCT applications, particularly in settings where resources are constrained.

## Figures and Tables

**Figure 1 sensors-23-08524-f001:**
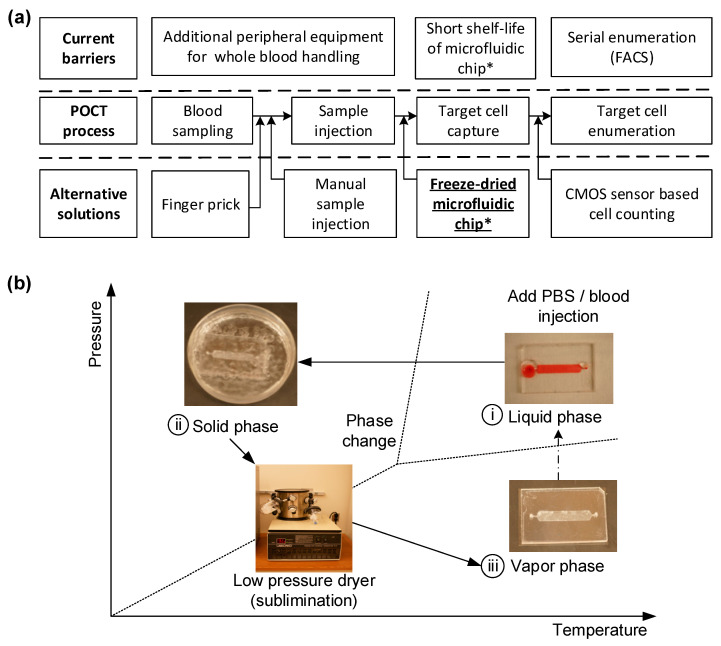
Overview of requirements and fabrication for a freeze-dried immunoassay microfluidic chip for point-of-care testing (POCT). (**a**) Overall steps for POCT. (**b**) Fabrication workflow: The freeze-drying process involves a sublimation phenomenon under low-pressure conditions to convert the liquid phase into a solid. (i) Immersing in liquid nitrogen, (ii) reducing ambient pressure, and (iii) initiating sublimation; the solid phase moves directly into the gaseous phase without transitioning through a liquid state (*, denotes substitutable device).

**Figure 2 sensors-23-08524-f002:**
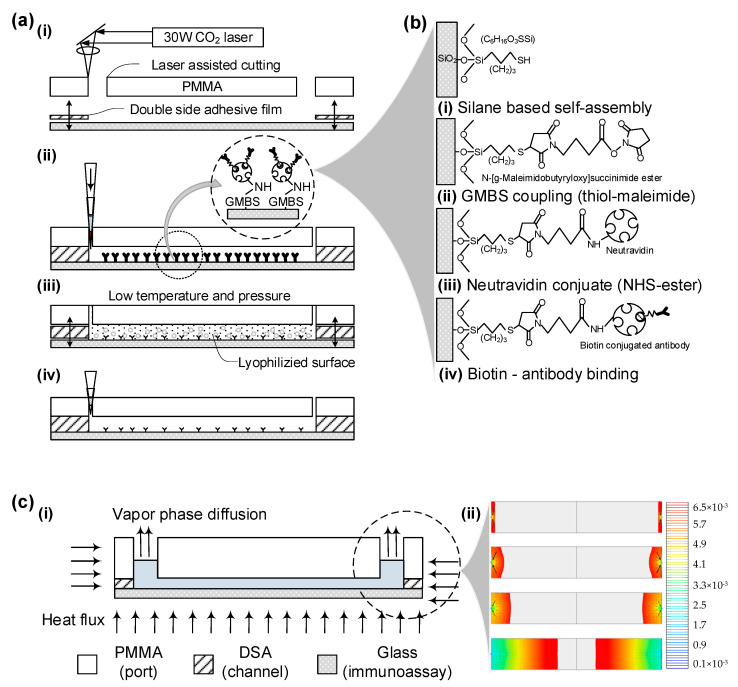
This figure presents a comprehensive analysis of the sublimation process in a fabricated microfluidic chip through numerical simulation and surface chemistry. (**a**) Shows the microfluidic channel geometry and parameters for sublimation, modeled using COMSOL Multiphysics^®^ software, V5.0. (**b**) Illustrates the surface chemistry techniques used to immobilize target antibodies onto a glass slide. (**c**) Provides an overview of the design and modeling parameters for the microfluidic system.

**Figure 3 sensors-23-08524-f003:**
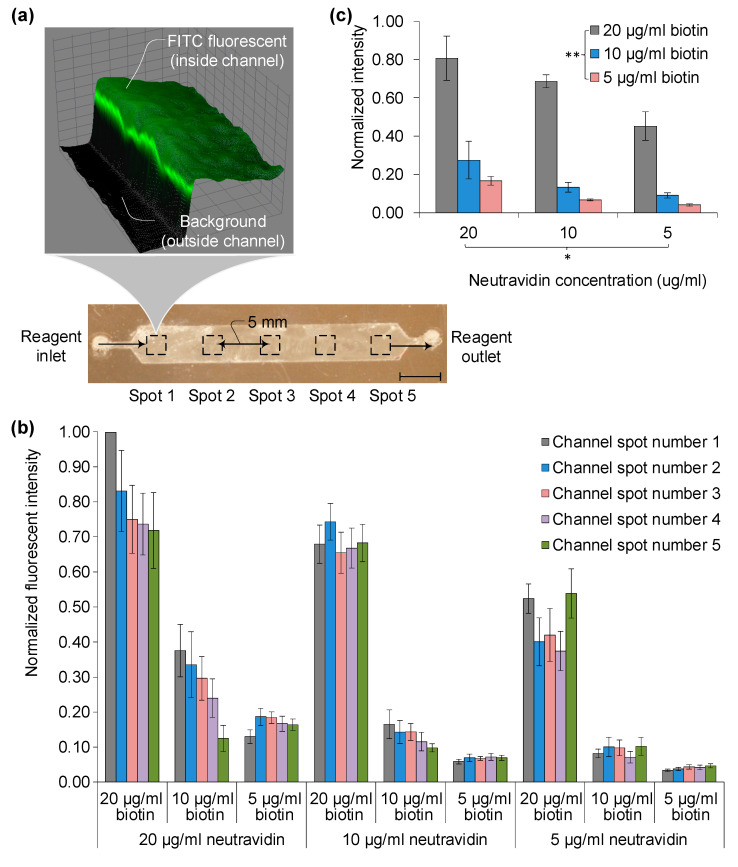
Surface functionality assessment via fluorescent tagging techniques and control surface comparison pre-freeze-drying. (**a**) Evaluation involves a modified surface chemistry protocol for immunoassay functionality, following neutravidin binding and biotinylated fluorescent tagging to measure binding specificity; scale bar denotes 5 mm. (**b**) Depiction of five distinct regions on the surface. (**c**) Presentation of green fluorescent intensity images alongside a graph comparing biotin concentrations. Statistical analysis using two-way ANOVA without replication indicates significant differences concerning two conjugation molecules and their concentrations: *p* * signifies a 0.04 significance level for neutravidin and *p* ** denotes a 0.001 significance level for biotin, falling below the 95% confidence threshold (α = 0.05).

**Figure 4 sensors-23-08524-f004:**
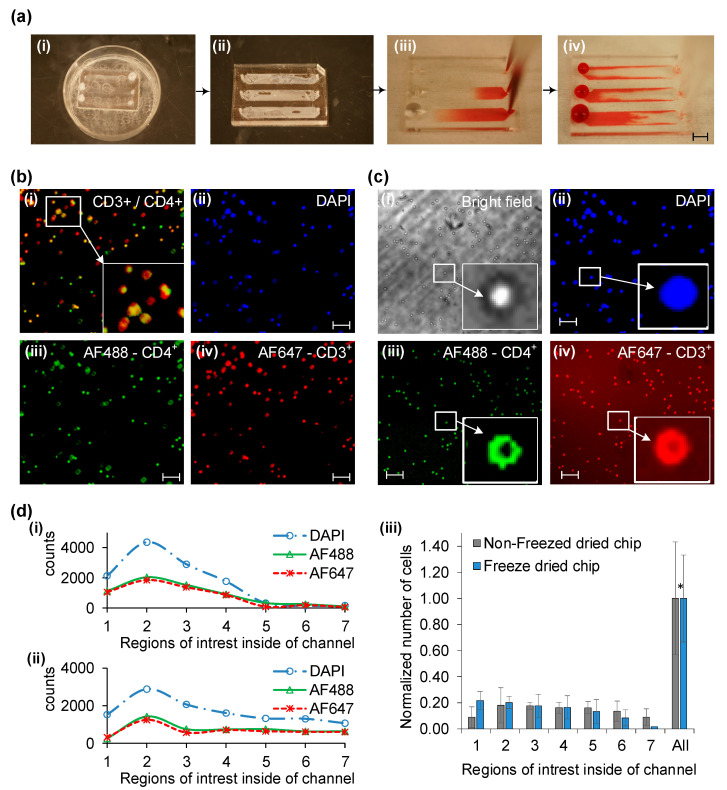
Comparing cell capture efficacy between a standard and freeze-dried microfluidic chip through whole blood injection, with assessment via fluorescent imaging and bright field cell counting. (**a**) Steps in utilizing the freeze-dried immunoassay-based microfluidic chip: first, the surface-treated chip is dipped in liquid nitrogen (i), followed by device reassembly (ii), whole blood injection (iii), and wash out (iv). Scale bar measures 5 mm. (**b**) Fluorescent imaging of captured cells after immunostaining on a standard chip includes overlay images for DAPI/AF488/AF647 (i). DAPI (ii), cell membrane antibody CD4^+^ with AF488 in green (iii), and CD3+ with AF647 in red (iv). (**c**) A set of fluorescent and bright field images (i) captured from the freeze-dried chip, showing cells labeled with DAPI (ii), AF488 (iii), and AF647 (iv). The enlarged image distinctly showcases the validity of three fluorescence staining: DAPI, AF488, and AF647. (**d**) Quantitative analysis of captured cells: cell count (i); freeze-dried chips (ii) using DAPI/AF488/AF647 markers; normalized cell counts (*n* = 6) (iii). Scale bars indicate 100 μm. The calculated *p*-values from a Student’s *t*-test based on two-sided distributions with equal variance show *p* * = 0.208, which is greater than 0.05, indicating a 95% confidence level.

**Figure 5 sensors-23-08524-f005:**
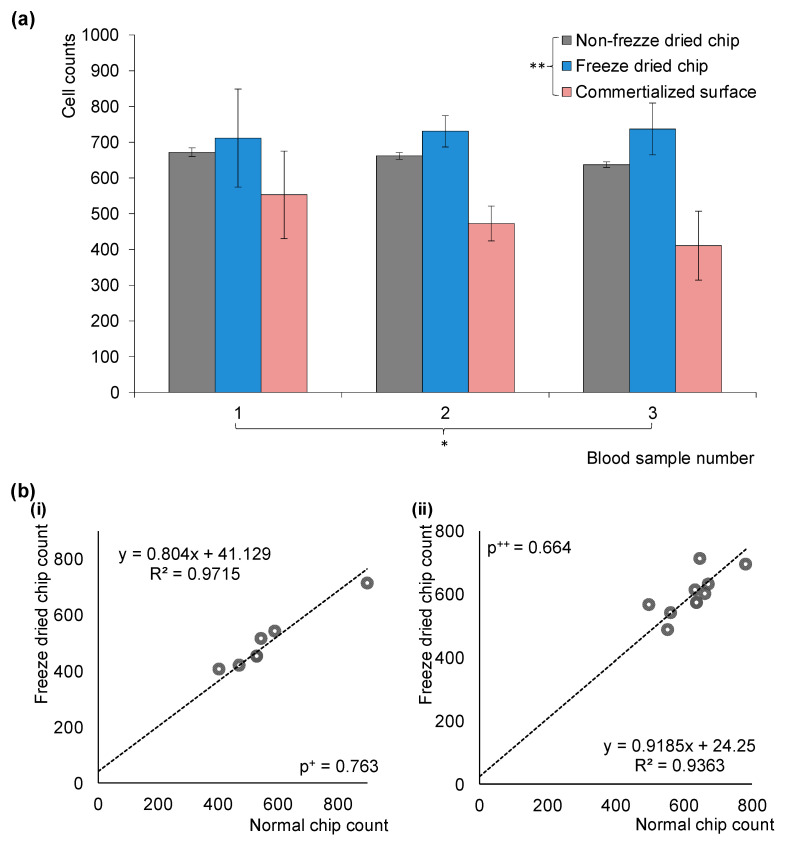
Comparing the efficiency of normal, freeze-dried, and commercially available non-specific protein chips in capturing cells from diverse blood samples. (**a**) Examination of cell capture efficiency across three chips with three different blood samples reveals no significant difference for normal blood samples (*p* * = 0.43), as confirmed via two-way ANOVA without replication. However, variances between chip types yield a lower *p*-value ** of 0.004, considered statistically significant at a 95% confidence level (α = 0.05). (**b**) Evaluation of correlation factors for different blood samples, using 12 microfluidic chips (i) and 18 microfluidic chips (ii), split evenly between normal and freeze-dried chips. Correlation factors of 0.97 and 0.93 were obtained for two different blood samples. Student’s *t*-test analyses, based on two-sided distributions with equal variance, indicate no significant difference between chip types for blood sample number 1 (*p*^+^ = 0.763) and blood sample number 2 (*p*^++^ = 0.664), both falling above the 95% confidence level (α = 0.05).

**Table 1 sensors-23-08524-t001:** Freeze-drying procedure for microfluidic device.

	Step	Methodology	Reaction Time and Temperature Condition
1	Blood Injection	Inject 5 μL of buffy coat blood of blood: serum concentration ratio (1:0, 1:1, 1:2)	10 min at low speed
2	Lysing solution	Inject 100 μL of lysing solution	

## Data Availability

Not applicable.

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
