# Peer review of "Extending the Shelf-Life of Immunoassay-Based Microfluidic Chips through Freeze-Drying Sublimation Techniques"

_sensors, 2023, doi:10.3390/s23208524_

Round 1

Reviewer 1 Report

The authors have studied different conditions that can be useful to the community. However, there need to be better quality data presentation in general.

1. Can you try different reagents because each one will have different shelf-life and properties?

2. Can you also include room temperature dry?

3. In Figure 1(a), you need to fix self-life to shelf-life.

4. In Figure 3(b), you need to add error bars.

5. In Figure 4(b)(c), I don't see any membrane staining. You should include zoomed-in images that clearly show membrane staining. 

6. Make all the bar graphs in different colors for different conditions, instead of different patterns, which is hard to recognize.

7. References are in a different font.

N/A

Author Response

Answers to the reviewer’s comments

I am re-submitting the enclosed material after revision for possible publication in your journal. We sincerely thank the reviewers for their careful reading of the manuscript and valuable comments. We have made revisions according to the reviewers’ helpful comments and suggestions, as described below. The revised portions of the manuscript are highlighted in blue.

The authors have studied different conditions that can be useful to the community. However, there need to be better quality data presentation in general.

  1. Can you try different reagents because each one will have different shelf-life and properties?

Ans.: Thank you for your insightful comments. Regarding the expiration dates of our reagents, we have adhered to the specified ranges provided by the manufacturers, with some reagents lasting between 6 months to 2 years (Ethanol, 3-MPS, DMSO, IPA) and others less than 1 year (PBS, NeutrAvidin, biotinylated antibody, Lysing solution). These are stored appropriately, either in liquid form or frozen, and are then prepared based on Table S1 when a surface fixation process is required. We acknowledge that the primary concern here is the potential characteristic changes during the fixation process. However, given that our fixation process is conducted within a brief 5-hour window, we anticipate minimal, if any, changes in surface properties.

For our experiments, we employed multiple incubation periods and found that, due to diffusion characteristics in the microfluidic chip and reagent concentrations, a fixation time of roughly 1 hour yielded the best results, and this approach was subsequently adopted. We recognize the importance of your point regarding potential characteristic changes related to reagent expiration dates. This is indeed an area we believe warrants further investigation in subsequent research.

Lastly, as shown in Figure 3(b) and 3(c), we illustrated the efficiency based on the channel position and Neutravidin concentration, hypothesizing the significant influence of Neutravidin on the biotinylated antibody. It is designed with the primary goal of maximizing surface fixation efficiency, but we concur that other reagents with distinct properties might require a separate set of experimental approaches to clarify their characteristics.

  1. Can you also include room temperature dry?

Ans.: We have presented the data under dry conditions in Figure 4d(iii). The indoor temperature was compared for two drying states within a standard "room temperature" environment.

  1. In Figure 1(a), you need to fix self-life to shelf-life.

Ans.: We have made the requested changes.

  1. In Figure 3(b), you need to add error bars.

Ans.: Thank you for bringing this to our attention. In Figure 3(b), we measured the efficiency by normalizing the fluorescence intensity of a combination of 20ug/ml Neutravidin and 20ug/ml biotin. Therefore, we have displayed the average and deviation for the entire fluidic chip, which includes location-specific characteristics, in Figure 3(c). We incorporated location-specific variability by adding error bars to represent the overall chip error and the normalization process.

  1. In Figure 4(b)(c), I don't see any membrane staining. You should include zoomed-in images that clearly show membrane staining.

Ans.: To ensure the membrane staining is clearly visible, we adjusted the sharpness and gamma values of the image. Additionally, we have included a magnified image in Figure 4(c) to distinguish cells where the three fluorescent stains overlap.

  1. Make all the bar graphs in different colors for different conditions, instead of different patterns, which is hard to recognize.

Ans.: Thank you for your observation. We have color-coded the patterns in the bar graph using distinct colors to enhance readability.

  1. References are in a different font.

Ans.: Thank you for pointing that out. We have changed the font of the references to maintain stylistic consistency throughout the document.

Reviewer 2 Report

The article proposed a freeze-dry sublimation process designed to mitigate the degradation of immobilized antibodies to extend the shelf-life of immunoassay based microfluidic chips.

The article can be considered for publication in "Sensors" after revising the following questions. The comments are below.

1) In introduction, the authors illustrated the defects of immunoassay based microfluidic chips and introduced freeze-dry sublimation. However, too many paragraphs in introduction will disrupt the logic of the article and confuse viewers. It is difficult for the audience to understand the article through the introduction. In introduction, it is highly recommended that the author consolidate some paragraphs into one paragraph.

2) In the introduction, it is advisable to include references to similar work and provide a critical review of simultaneous studies in the same field. This addition will enhance the comprehensiveness of the introduction.

3) In the introduction, the authors describe the main materials for immunoassay based microfluidic chips, including PMMA, glass slides, and double-sided adhesive films. PDMS, as one of the main materials for microfluidic chips, should be introduced for its application in immunity.

Analyst, DOI: https://doi.org/10.1039/C5AN01835H

Lab on a chip, DOI: https://doi.org/10.1039/C7LC00249A

Journal of Materials Chemistry B, DOI: http://doi.org/10.1039/d2tb02338e

4) In figure 1a, μ-fluidic chip is an uncommon usage. Microfluidic or nanofluidic is highly recommended.

5) Starting from page 5, line numbers are missing. It is recommended to correct this oversight.

6) In figure 1b, the authors introduced the fabrication workflow. I suggest changing the representation of the diagram. The current chart looks confusing.

7) The illustrations of figures 1 and 4 are too long to understand. If the authors tend to introduce the content of figures, I recommend adding extra description to the text instead of illustrations.

8) In figures 2 and S1, the units of the simulated parameters should be marked in the figure. Subsequently, the COMSOL simulation calculation formula and related data should be provided.

9) Microfluidic chip is a very important component, the author should provide more dimensions of the chip, if necessary, should be marked in the figure.

10) How does the cost of using freeze-dried chips compare with ordinary chips? If the cost is too high, the popularity of this method will be affected.

11) In figure 3b, are there differences in the parameters from Channel spot1 to 5? If not, please explain why there is a difference in fluorescence for the same concentration of biotin in figure 3b or calculate RSD for the same group.

12) The blank from page 7 should be adjusted.

13) In the conclusion section, including some limitations of the study and proposing directions for future research would enhance the paper's professionalism.

Author Response

Answers to the reviewer’s comments

I am re-submitting the enclosed material after revision for possible publication in your journal. We sincerely thank the reviewers for their careful reading of the manuscript and valuable comments. We have made revisions according to the reviewers’ helpful comments and suggestions, as described below. The revised portions of the manuscript are highlighted in blue.

The article can be considered for publication in "Sensors" after revising the following questions. The comments are below.

  1. In introduction, the authors illustrated the defects of immunoassay based microfluidic chips and introduced freeze-dry sublimation. However, too many paragraphs in introduction will disrupt the logic of the article and confuse viewers. It is difficult for the audience to understand the article through the introduction. In introduction, it is highly recommended that the author consolidate some paragraphs into one paragraph.

Ans.: Thank you for your feedback. To address your concerns, we have integrated the paragraphs beginning at line 77 and line 85 into one, as they both present solutions to the issues raised in our paper. Additionally, the paragraph starting at line 110 has been merged for continuity, as it offers solutions pertaining to the POC requirements discussed in the previous section. The paragraph beginning at line 143, which discusses the application of the current POC method to HIV testing, has been combined with the subsequent paragraph in the introduction to enhance clarity and understanding.

  1. In the introduction, it is advisable to include references to similar work and provide a critical review of simultaneous studies in the same field. This addition will enhance the comprehensiveness of the introduction.

Ans.: We have referenced studies in the same field from lines 74-77 to present the relevant area and subsequently outlined its limitations after line 77. Specifically, we've revised our critical review of the dry chip method at line 85 as follows. In the following sections, we elaborated on solutions to overcome these limitations and described approaches for application in the same domain.

(Line : 85-91).

While this approach leverages surface fixation to matrices, applying it directly to the sensitive planar micro-fixation technique aimed at detecting specific cells within blood can be challenging. Furthermore, by using alternative reagents based on sugars like trehalose, it necessitates modifying the surface using a cleaning solution before introducing the blood sample. These additional steps can impact the capture efficiency of specific cells, demanding preservation methods for equipment at the end-user stage.

  1. In the introduction, the authors describe the main materials for immunoassay based microfluidic chips, including PMMA, glass slides, and double-sided adhesive films. PDMS, as one of the main materials for microfluidic chips, should be introduced for its application in immunity.

Analyst, DOI: https://doi.org/10.1039/C5AN01835H

Lab on a chip, DOI: https://doi.org/10.1039/C7LC00249A

Journal of Materials Chemistry B, DOI: http://doi.org/10.1039/d2tb02338e

Ans.: Thank you for your suggestion. The three additional references you provided propose manufacturing methods for microfluidics using polymer materials, including PDMS. We have incorporated these three papers into the introduction and updated our references accordingly.

  1. In figure 1a, μ-fluidic chip is an uncommon usage. Microfluidic or nanofluidic is highly recommended.

Ans.: Thank you for pointing that out. We have replaced the non-standard terminology in Figure 1(a) with the term 'microfluidics'.

  1. Starting from page 5, line numbers are missing. It is recommended to correct this oversight.

Ans.: Due to an error during the conversion process, we have now inserted line numbers throughout the entire document.

  1. In figure 1b, the authors introduced the fabrication workflow. I suggest changing the representation of the diagram. The current chart looks confusing.

Ans.: The diagram summarizes the sublimation process, and we have revised any unnecessary expressions.

  1. The illustrations of figures 1 and 4 are too long to understand. If the authors tend to introduce the content of figures, I recommend adding extra description to the text instead of illustrations.

Ans.: Thank you for your suggestion. We have simplified the descriptions for Figures 1 and 4 and made the following adjustments to the main text accordingly.

(Line : 124-134).

To extend the shelf-life of a microfluidic chip designed for POCT, a freeze-drying technique is applied. The chip, initially functionalized in a liquid environment for immunoassay procedures, needs to be freeze-dried to maintain its effectiveness until used for blood testing. Fig. 1(b) shows fabrication Workflow: The freeze-drying process involves a sublimation phenomenon under low-pressure conditions to convert the liquid phase into a solid. These steps include: (i) Immersing the antibody-immobilized microfluidic chip in liquid nitrogen to solidify the buffer solution; (ii) Reducing ambient pressure while maintaining low temperature, without changing the phase; (iii) Initiating sublimation by incrementally increasing pressure, thus turning the solid phase directly into the gaseous phase without transitioning through a liquid state.

  1. In figures 2 and S1, the units of the simulated parameters should be marked in the figure. Subsequently, the COMSOL simulation calculation formula and related data should be provided.

Ans.: We have included the parameters and units used in the supplementary figure. Additionally, based on Section 2.1, we have provided the formulas used in the simulation in tabular form.

  1. Microfluidic chip is a very important component, the author should provide more dimensions of the chip, if necessary, should be marked in the figure.

Ans.: Thank you for bringing this to our attention. Through the scale bar accompanying the figure, one can discern the planar dimensions of the chip, and the thickness dimension of the materials used can be inferred from Section 2.2. For clarity, we have included a figure with dimensions in the supplementary figure.

  1. How does the cost of using freeze-dried chips compare with ordinary chips? If the cost is too high, the popularity of this method will be affected.

Ans.: The fabrication process of the freeze-drying chip involves two additional steps: cooling with liquid nitrogen and low-pressure drying. While these extra processes may increase the cost compared to a standard chip, employing a batch process to freeze-dry a large quantity of chips simultaneously can offset these expenses. Moreover, the only material needed for this process is liquid nitrogen, so we anticipate minimal additional costs related to materials. However, considering the costs associated with logistics for standard chips and limitations in shelf life that could raise storage costs, we believe that this can be a viable option for mass production in product development.

  1. In figure 3b, are there differences in the parameters from Channel spot1 to 5? If not, please explain why there is a difference in fluorescence for the same concentration of biotin in figure 3b or calculate RSD for the same group.

Ans.: Thank you for pointing this out. The differences in fluorescence observed between spots can be attributed to the surface forces arising from shear stress distribution in the fluid with laminar flow and to the diffusion variables (both temporal and spatial) from physical and chemical absorption during chemical reactions. As such, we have represented the relative differences concerning the standard surface absorption amount (based on 20ug/ml) in the result graphs. As shown in Figures 3b and 3c, it's evident that the fluorescence differences between spots are more closely related to the concentration of the reagent attached to the surface. We've updated the explanation accordingly, as reflected on line 370.

(Line : 370-373).

Furthermore, the differences in fluorescence occurring at each spot can be attributed to the surface forces resulting from shear stress distribution in the fluid where laminar flow occurs, as well as the variables of diffusion (both temporal and spatial) from physical and chemical absorption during chemical reactions.

  1. The blank from page 7 should be adjusted.

Ans.: We observed blank spaces due to paragraph spacing adjustments in Table 1 and have made the necessary edits to rectify it.

  1. In the conclusion section, including some limitations of the study and proposing directions for future research would enhance the paper's professionalism.

Ans.: Thank you for highlighting this aspect. We discussed the limitations of our study from line 493, covering the generation of salt during the rehydration process, efficiency changes due to manual injection, and automated imaging solutions for cell detection. We've proposed the incorporation of a double wash step, the use of auto-pipettes, and the introduction of in-line holographic imaging as potential directions for future research. We are currently continuing our investigations in line with these suggested research directions.

Round 2

Reviewer 1 Report

The author addressed all the comments. However, I'm still not clear about Figure 4b. Why does DAPI have ring structures? Also, why do CD3+ and CD4+ cells do not show ring structures? Even though the setting was adjusted, I only see blobs instead of a clear, ring-like membrane staining. The author has to redo the experiment to achieve correct staining patterns.

NA

Author Response

Answers to the reviewer’s comments

We've made some updates to the attached content and are hoping for it to be considered for your journal. Our manuscript has greatly benefited from the feedback of our peers. We've made the necessary changes based on their input. You can find the edited sections highlighted in blue.

The author addressed all the comments. However, I'm still not clear about Figure 4b. Why does DAPI have ring structures? Also, why do CD3+ and CD4+ cells do not show ring structures? Even though the setting was adjusted, I only see blobs instead of a clear, ring-like membrane staining. The author has to redo the experiment to achieve correct staining patterns.

Ans.: Thank you for your insightful comments. As shown in Ref Figure 1 below, cells trapped on the surface in the bright field are stained with DAPI, indicating that they are a type of white blood cell containing a nucleus. This staining is done to confirm that the trapped particles are cells. Additionally, images from Ref Figures 2 to 4 are used to determine whether the trapped cells at the same location are CD4+, CD3+, or CD3+/CD4+. As seen in the provided images, the nuclear staining pattern, DAPI, displays a solid fluorescence image, while CD3+ and CD4+ are stained on the cell membrane, showing fluorescence brightness concentrated around a ring. However, such fluorescent images are distinctly noticeable in high magnification images, but the differences are not clear in low magnification images for purposes like cell counting in this paper. This is because, in a 3D cellular context, the fluorescent image at low magnification overlaps the cell nucleus and the cell membrane on the same plane. Therefore, after verifying the correct staining pattern through the reference images below, it was utilized for cell counting in low magnification images.

Ref Figure1. Bright field and DAPI staining combination:

Ref Figure2. Nuclei staining Blue

Ref Figure3. Membrane staining Green

Ref Figure4. Membrane staining Red

Reviewer 2 Report

no comments.

Author Response

Answers to the reviewer’s comments

We've made some updates to the attached content and are hoping for it to be considered for your journal. Our manuscript has greatly benefited from the feedback of our peers. We've made the necessary changes based on their input. You can find the edited sections highlighted in blue.

The author addressed all the comments.

Ans.: Thank you for your insightful comments.